# Effect of the Localized Insecticides Spray Technique to Control *Halyomorpha halys* in *Actinidia chinensis* Orchards

**DOI:** 10.3390/insects15090723

**Published:** 2024-09-20

**Authors:** Matteo Landi, Michele Preti, Antonio Masetti, Francesco Spinelli

**Affiliations:** 1ASTRA Innovazione e Sviluppo, 48018 Faenza, Italy; 2Dipartimento di Scienze e Tecnologie Agro-Alimentari, Università di Bologna, 40127 Bologna, Italy; antonio.masetti@unibo.it

**Keywords:** brown marmorated stink bug, kiwifruit, integrated pest management, fruit damage, chemical control, atomizer, insecticide application

## Abstract

**Simple Summary:**

*Halyomorpha halys*, the brown marmorated stink bug, is an invasive insect pest that in the last two decades became a major problem for several agricultural commodities, including kiwifruit. Its management relies on integrating different control methods, including board-spectrum insecticides. Nevertheless, chemical control measures may not achieve satisfactory results and there is still margin for improvement, considering for instance the optimization of the spray technique. This problem is even more relevant in kiwifruit due to its peculiar training system, which reduce fruit exposure to insecticide sprays. In this study, two spray techniques were tested to maximize the insecticides efficacy in controlling *H. halys* in both yellow- and green-flesh kiwifruit cultivars (‘Jintao’ and ‘Hayward’). The effects of a conventional ray atomizer and a trumpet-modified atomizer, which localizes insecticide applications in the fruit area, were assessed on *H. halys* mortality (with artificial infestations) and fruit damage (due to *H. halys* naturally occurring in the orchards). The localized spray technique resulted in an overall significantly higher *H. halys* mortality in ‘Hayward’, but not in ‘Jintao’ cultivar. This is likely due to differences in the canopy size and structure of these cultivars. However, the fruit injury level was not different between the spray techniques. Further investigations in this direction are needed also to assess the efficiency of localized spray technique in reducing the insecticide rates, in accordance with European strategies that foreseen restrictions in the authorized active substances usage.

**Abstract:**

Kiwifruit (*Actinidia chinensis*) cultivation is expanding worldwide, with China, New Zealand, and Italy being the major producing countries. *Halyomorpha halys*, the brown marmorated stink bug, is raising serious concerns to kiwifruit cultivation both in China and Italy. This study aimed at improving the chemical control efficacy against this pest by comparing two insecticide spray techniques (a conventional ray atomizer and a trumpet-modified atomizer adapted for localized spray application) in kiwifruit. In fact, kiwifruit is often grown with a ‘pergola’ training system, which may reduce the effectiveness of insecticide penetration into the canopy. Experiments were performed in naturally infested orchards of both *Actinidia chinensis* var. *chinensis* ‘Jintao’ and *A. chinensis* var. *deliciosa* ‘Hayward’. Furthermore, mesh cages containing *H. halys* adults were deployed within orchards to assess the insects’ mortality at 1, 3, 7, and 10 days after an insecticide application with two spray techniques during two consecutive seasons. In the cultivar ‘Jintao’, the two systems performed similarly, while in the cultivar ‘Hayward’, an overall significantly higher insect mortality was recorded with the trumpet atomizer (94–100%) compared to the conventional atomizer (59–78%). Crop damage was also evaluated on both cultivars, simulating the grower insecticide applications with the two spray techniques. At harvest, no difference emerged between the spray techniques, which provided a significantly better protection compared to the untreated control (12–17% compared to 33–47% of injured fruits). Further investigations in this direction are needed also considering the restriction of insecticidal active substances ongoing in the European Union and the need to maximize the efficacy of the available tools.

## 1. Introduction

Kiwifruit, *Actinidia chinensis* (Planch., 1847), originated in China and it was domesticated in New Zealand from where it became a global crop [1]. The kiwifruit market is dominated by two major varieties: the green-flesh (*A. chinensis* var. *deliciosa*) and yellow-flesh cultivars (*A. chinensis* var. *chinensis*) [2,3]. Despite kiwifruit cultivation being very successful and widely spread, with an increase in total fresh fruit production from 577 to 887 Mt/year in the last 20 years [4,5], this crop is facing several phytoiatric problems. Among them, *Pseudomonas syringae* pv. *actinidiae* [6], kiwifruit vine decline syndrome, KVDS [7], and *Halyomorpha halys* [8] pose the highest risks.

In Italy, more than 45% of kiwifruit production is concentrated in Lazio (specifically in Agropontino, a Latina province), followed by Emilia-Romagna (in particular Ravenna province), which accounts for 15% of national production. Among the other Italian regions, the most important are Veneto and Piedmont (with approximately 10% each of the Italian kiwifruit production), as well as Calabria and Campania (together with another 13% of the total national production) [9]. The Italian kiwifruit production covers a crucial economic importance for both the national market and international export, mainly in Europe [10]. Given the value generated by this crop for the Italian growers and for the whole kiwifruit industry, it is necessary to guarantee high quality standards, while minimizing the inputs of plant protection products [11]. In 2020, China produced about 50% of the global kiwifruit production, whereas New Zealand and Italy accounted for 14% and 12%, respectively [12]. Prior to 2020, Italy ranked second in total kiwifruit production, ahead of New Zealand, and this shift in global ranking could be related to the adverse climatic conditions registered in Italy in the recent years and to the rise yield losses due to several factors, including KVDS and *H. halys*.

The brown marmorated stink bug, *Halyomorpha halys* (Stål) (Hemiptera: Pentatomidae) is a very concerning invasive insect species that caused important economic losses in most countries outside its Asian native range [13,14,15,16]. Major damage was reported mainly on tree fruit crops, including kiwifruit [8,17,18,19]. This pest was introduced in Europe in the early 2000s, likely with international trades. In Italy, the first record dates back to 2007 in Liguria [20], but about ten years ago, the first region where economic losses were reported was Emilia-Romagna [21,22]. In kiwifruit crop, the feeding punctures of *H. halys* cause internal white corking damage to the fruit flesh, which is visible only in peeling the fruit skin [23,24], and in some cultivars the fruit injuries can lead to an early drop [25]. *Halyomorpha halys* is bivoltine in Italy, and both generations occur in Italian kiwifruit orchards [8]. The highest densities of overwintered adults usually occur at the end of kiwifruit flowering (crop BBCH 69), the first generation adults are recorded during the late fruit development (crop BBCH 79), and the second generation adults around harvest (crop BBCH 85). Nymphs appear in late spring and occur until late summer. Both nymphs and adults can cause severe damage to kiwifruit during the whole period ranging between fruit set and harvest [8].

*Halyomorpha halys* management requires the adoption of several complementary control methods to be applied with an integrated approach [16]. The first pillar is to create an ecological equilibrium, implementing the biological control, mainly carried out by egg parasitoids [26,27,28] that were recorded also within kiwifruit orchards [8,29]. The use of exclusion netting systems also proved to be effective in a number of crops towards several insect pests, including *H. halys* [30,31,32], and it can reduce fruit damage also in kiwifruit orchards [33]. Then, the practical implementation of chemical ecology knowledge, exploiting for instance synthetic *H. halys* aggregation pheromones [34], led to the creation of monitoring networks at a regional scale with a shared alert system, the issuance of weekly bulletins, and the development of a forecasting model in order to properly time the insecticide applications against this pest [35].

However, chemical control of *H. halys* using insecticides remains an indispensable and crucial component for the successful management of this pest. The achievement of a satisfactory chemical control of *H. halys* is very challenging because of the characteristics of this pest (e.g., polyphagia, mobility across different crops, and recovery ability when exposed to sublethal doses of insecticides) and the features of available effective active substances (e.g., short-time residual activity, lack of selectivity towards non-targets) [36,37,38,39,40,41]. Organic insecticides show some promising results under laboratory conditions [42], but insecticides authorized on kiwifruit crop in organic farming (e.g., pyrethrins) do not match the performances of synthetic insecticides (e.g., etofenprox, deltamethrin) [43]. In addition, the European legislation is moving toward the reduction in synthetic plant protection products, with the ban of several active substances and numerous limitations in the use of the authorized ones [44,45]. Therefore, the optimization of all available management practices for *H. halys*, considering the above-mentioned factors, is crucial.

Especially the application of insecticides needs to be carefully rationalized and optimized, aiming at coupling the maximum efficacy with the best cost-effective convenience. First, all the spraying machineries require a regular calibration in order to fit the operative parameters to the specific field conditions, aiming at providing a uniform, effective, reliable, and targeted distribution of the plant protection products in each crop system. Then, the choice of the active substance (a.s.) as well as the proper timing play a major role on the performance of any insecticide application. In addition to that, from a sustainability standpoint, also the environmental and social impacts need to be properly evaluated in order to guarantee the lowest possible impact of these practices on the agroecosystems and the minimum possible residues on fruit at harvest. Furthermore, due to the peculiar training systems used on vines, plant protection products distribution in kiwifruit orchards is challenging. Indeed, kiwifruit vines are generally trained in pergola or T-bar trellis systems [46], and in these conditions canopy size can reach 50,000 m^2^/ha [47]. Additionally, the pergola system hampers air circulation, increases humidity inside the canopy, and may reduce the efficient penetration of the plant protection products. In this context, the technique of the insecticide distribution within the crop canopy and the quantity reaching the target areas (i.e., where the fruits are localized, to protect them from the *H. halys* feeding activity) are crucial factors.

This study was aimed at comparing two insecticide spray techniques to control *H. halys* in two kiwifruit cultivars in Emilia-Romagna, Northern Italy. The specific objectives were to:(i)evaluate the mortality over time caused by pyrethrins/pyrethroids applied either with a conventional ray atomizer or a localized spray technique on *H. halys* adults;(ii)assess whether the different spray techniques had any effects on fruit damage caused by *H. halys* at harvest, considering the same grower insecticide spray program.

## 2. Materials and Methods

### 2.1. Spraying Machineries

In this study, the conventional ray atomizer (Figure 1A) was compared to a localized spray technique where the insecticide product is delivered by means of trumpets that restrict the spray range in the area where fruits are located (Figure 1D). The trumpets-modified atomizer does not differ in the body structure from the conventional ray atomizer, sharing the chassis and all the electronical and mechanical components. The difference between these spray machineries is in the area where the air flow exits and in the position of the nozzles; therefore, in the delivery trajectory of the plant protection products. In the localized spray, there is a metal deflector positioned in the area from which the air generated by the fan exits, conveying it through plastic pipes on which end nozzles are mounted. In the trumpets-modified atomizer, the nozzles are mounted on a nozzle holder positioned on the side of the fan and connected to the frame via a metal structure, forming a so-called ‘trumpet’ (Figure 1E).

The trumpets-modified atomizer is usually utilized by kiwifruit growers for artificial pollination during blooming, providing a localized air flow in the area of the canopy where most of the flowers occur. In all the experiments, the same atomizer (model Vulcano Georgia 2000, Vulcano srl, Faenza, RA, Italy) was used, considering the conventional application technique when using the full ray operative (with 10 nozzles, model ATR Albuz 80, of which 2 were green, 6 grey, and 2 red, for a nominal flow rate of 29.5 L min^−1^) and the localized system technique when the trumpets were installed on the atomizer (with 8 nozzles model ATR Albuz 80, all green, for a nominal flow rate of 27.4 L min^−1^). The same operative parameters (operational pressure 20 atm, speed 7.5–8 km h^−1^, and distributed water volume 800 L ha^−1^) were kept in each application and in all the experiments for both the insecticide spray techniques.

### 2.2. Study Sites and Biometric Plant Parameters

Experiments were carried out during 2021 and 2022 simultaneously in the yellow-flesh and the green-flesh kiwifruit cultivars ‘Jintao’ and ‘Hayward’, respectively. All the experiments were carried out in a farm located in Castel Bolognese (RA) (44°19′36.74′′ N, 11°47′38.17′′ E), where two neighboring commercial kiwifruit orchards (<20 m apart) were managed by the same grower following the Good Agricultural Practices (GAP) and according to the Integrated Pest Management (IPM) guidelines. In the area, *H. halys* infestations have naturally occurred every year since 2015 [8,22].

The orchard of the cultivar ‘Jintao’ (0.5 ha) was planted in 2007 with a planting space of 4.5 m between rows and 1.5 m between plants (density: 1481 plants ha^−1^). The orchard of the cultivar ‘Hayward’ (1.2 ha) was planted in 2018 with a planting space of 4.6 m between rows and 2.0 m between plants (density: 1087 plants ha^−1^). Both cultivars were grafted on ‘Hayward’ rootstock and the orchards had the same density of male plants (1 every 5 female plants). In both orchards, the pergola training system was used and vines were irrigated by a drip system. The ‘Jintao’ orchard had an anti-hail net cover combined with lateral nettings (i.e., monoblock netting system) to protect the crop from both hail and insect pests such as *H. halys*, while no nets were installed in the ‘Hayward’ orchard. The monoblock netting system is usually deployed in this orchard cultivar ‘Jintao’ after blooming, at the end of May.

Despite the same training system being used, the differences between the cultivars ‘Jintao’ and ‘Hayward’ in terms of vigor and pruning technique affect the canopy structure and density, and could impact the outcomes of insecticide applications performed with different spray techniques (Figure 2). In particular, the green-flesh cultivars are usually pruned, leaving a higher number of productive shoots and longer shoots compared to the yellow-flesh cultivars. Therefore, the main biometric plant parameters of the two cultivars under study were recorded at the end of September 2021 (crop BBCH 79-81). Briefly, a random sample of 10 representative plants per each cultivar was selected within each orchard to count the number of productive shoots per plant. The length of all the shoots per plant was measured, as well as the total number of leaves in each shoot was counted; then, in one randomly selected shoot per each of the 10 plants, all the leaves were collected and scanned to measure the leaf area by means of the image processing software ImageJ [48].

### 2.3. Halyomorpha Halys Mortality Experiments

*Halyomorpha halys* individuals used for these experiments were field collected in unsprayed agroecological contexts and were previously kept in controlled conditions in the entomological laboratory of ASTRA Innovazione e Sviluppo (Faenza, RA, Italy), where they were fed with fresh organic vegetables (carrots, green beans, and tomatoes), sunflower and soybean seeds, and a cotton ball soaked with water. *Halyomorpha halys* individuals were held at a 12:12 light:dark photoperiod, a temperature of 25 ± 1 °C with a 60 ± 10% relative humidity prior to the field exposure in the mortality experiments.

To assess the insect mortality in function of the spray techniques, namely a conventional ray and trumpets-modified atomizer, an artificial infestation procedure was performed during the 2021–2022 seasons. The experiments were conducted following the methods described in Preti et al. [43] and according to the EPPO guidelines [49]. Three *H. halys* adults (sex ratio male:female = 1:2) were placed in mesh cages (20 cm × 30 cm, mesh size < 0.5 mm), which were wrapped around a single kiwifruit.

Each experiment was carried out following a block design with three replicates. Ten mesh cages were installed approximately in the center of each plot (30 *H. halys* individuals per replicate and 90 individuals per treatment). Cages were deployed on different plants and randomly selecting fruits with a variable degree of canopy coverage to simulate the natural pest occurrence within the orchard. Each experimental plot comprised a minimum of 4 neighboring rows and 10 consecutive plants within each row, for a minimum area of approximately 350 m^2^. All plots were located in the central part of the orchard at a minimum distance of 50 m from the borders in order to avoid perimeter-driven effects. Plots were at least 25 m apart from each other. The insect mortality was assessed at 1, 3, 7, and 10 days after the insecticide sprays, recording the number of alive and dead *H. halys* individuals in each mesh cage. The natural mortality was assessed, exposing groups of three *H. halys* adults in the field (within mesh cages) in plots unsprayed with insecticides (three control plots per experiment).

The mortality study considered a total of six replicated experiments (three per each kiwifruit cultivar). A single insecticide application was carried out per each experiment, using either the commercial product Asset Five^®^ (a.s. pyrethrins 46.5 g L^−1^) applied at 0.96 L ha^−1^ or Decis Evo^®^ (a.s. deltamethrin 25 g L^−1^) applied at 0.50 L ha ^−1^. In the cultivar ‘Jintao’, experiment #1 was carried out with deltamethrin on 30 August 2021 (crop BBCH 77-78), experiment #2 was carried out with pyrethrins on 21 September 2021 (crop BBCH 79-81), while experiment #3 was carried out with deltamethrin on 26 September 2022 (crop BBCH 79-81). In the cultivar ‘Hayward’, experiment #4 was carried out with pyrethrins on 21 September 2021 (crop BBCH 78-79), experiment #5 was carried out with deltamethrin on 15 October 2021 (crop BBCH 79-81), while experiment #6 was carried out with deltamethrin on 26 September 2022 (crop BBCH 78-79). Insecticide application timings and active ingredient choice replicated what Italian kiwifruit growers usually do to control *H. halys* infestations, following the products labels.

### 2.4. Injury Assessment on Kiwifruit

The impact of the insecticide application technique on the crop damage was measured on kiwifruits at commercial harvest in 2021. The cultivars ‘Jintao’ and ‘Hayward’ were managed following the same insecticide spray program. The two spray techniques (standard ray atomizer and trumpets-modified atomizer) were compared with an unsprayed control. The same plots used to test the insecticide efficacy with the artificial infestation of *H. halys* in mesh cages were sampled to score fruit damage caused by *H. halys* individuals naturally occurring in the study orchards. For the cultivar ‘Jintao’, the two insecticide applications were carried out in August and September 2021 with deltamethrin and pyrethrins, respectively, whereas for the cultivar ‘Hayward’, pyrethrins and deltamethrin were sprayed in September and October 2021, respectively (see experiments #1–2 for cultivar ‘Jintao’ and experiments #4–5 for cultivar ‘Hayward’).

At commercial harvest (crop BBCH 85-87), on 19 October 2021 for ‘Jintao’ and on 30 October 2021 for ‘Hayward’, samples of 100 randomly selected fruits per replicate were collected to assess the fruit injury level (300 fruits per treatment). These fruits were not collected from the same plants where the mesh cages were installed for the mortality study, previously flagged to be excluded from this study. After a 2-month period of cold storage at 4 ± 1 °C, each fruit was completely peeled to detect any internal white corking due to the *H. halys* feeding punctures and the number of injuries were recorded per each fruit (Figure 3). The fruit injury incidence was calculated as percentage of damaged fruits out of the total number of observed fruits, while the fruit injury severity was calculated as mean number of feeding injuries (i.e., internal white corking sites) per fruits considering all the observed fruits.

Finally, a monitoring trap (model Dead-Inn^®^ pyramid, AgBio Inc., Westminster, CO, USA) baited with *H. halys* aggregation pheromones (model Pherocon^®^ BMSB lure, Trécé Inc., Adair, OK, USA) was placed at approximately 100 m from the two kiwifruit orchards to record the *H. halys* population dynamics. The trap was checked from mid-May (crop BBCH 69-71) to the end of October 2021 (crop BBCH 85-87). The number of *H. halys* catches, separating adults and nymphs, was recorded weekly removing and killing all the caught individuals (Figure 4).

### 2.5. Statistical Analyses

Generalized linear mixed models (GLMM) with a binomial error distribution, a probit link function, and an autoregressive covariance structure of the first type were used to test the effects of treatments (conventional atomizer, trumpet atomizer, and untreated control) on the mortality of *H. halys* adults exposed within mesh cages. Days of exposure of the insects to the treatments were included as repeated measures, and the interaction “treatment × days of exposure” was tested as well. Mesh cages were nested within blocks and the Kenward–Roger method was used to estimate the degrees of freedom. Sequential Bonferroni methods were used to compare the levels of factors with significant effects setting the *p* level at 0.05. Separate models were run for each kiwifruit cultivar (‘Jintao’ and ‘Hayward’) and for each experiment because different insecticides were sprayed and the field trials were carried out in different periods of the seasons.

A factorial ANOVA model was used to test the effect of treatments (conventional atomizer, trumpet atomizer and untreated control) and the kiwifruit cultivar (‘Jintao’ and ‘Hayward’) on the mean number of *H. halys* feeding punctures per fruit. The interaction “treatment × cultivar“ was tested as well. Raw data were log transformed before running the model to match ANOVA assumptions and the Tukey HSD test was carried out for multiple comparisons of the three levels of the treatment factor setting the *p* level at 0.05.

The biometric plant parameters are reported as descriptive statistic, considering the mean values ± standard deviation.

The software package IBM SPSS Statistics (ver. 26) was used for all the analyses and for graphical representations of data.

## 3. Results

### 3.1. Effect of Insecticide Spary Technique on Halyomorpha halys Mortality in Conditions of Artificial Infestation

In experiment #1 carried out on the cultivar ‘Jintao’, the mortality of *H. halys* individuals assigned to the control groups drastically increased 3 days after the insecticide application. This result, which was likely due to a spell of very high temperatures recorded between the end of August and the beginning of September 2021, hampered the data analysis (Table 1). Therefore, only descriptive statistics for the assessment carried out 1 day after the application of deltamethrin are reported for experiment #1 (Table 2). 

In the two remaining experiments on the cultivar ‘Jintao’, a significant difference between the spray techniques was observed only at 1 and 3 days after insecticide applications (Table 1 and Table 2). With the pyrethrins-based organic insecticide, the conventional ray atomizer performed better than the trumpets-modified atomizer, and while using the synthetic pyrethroid deltamethrin, the trumpets-modified atomizer provided the best performance in terms of knock-down effect. One week after the application, both treatments provided a comparable efficacy, with a *H. halys* mortality significantly higher compared to that recorded in the untreated control (Table 2). The same trend was confirmed at the final assessment, when the overall mortality (considering together experiments #2 and #3) was 2.6-fold higher than what was recorded in the untreated control and no significant differences arose between the spray techniques.

In two out of three experiments carried out on the cultivar ‘Hayward’, the trumpet atomizer showed a significantly higher *H. halys* mortality than the conventional one (Table 1 and Table 3).

When the pyrethrins-based insecticide was applied on the cultivar ‘Hayward’ (experiment #4), the two spray technique results were comparable irrespectively from exposure intervals and different from the untreated control, with a final total mortality 1.7-fold higher than the unsprayed plots. In the experiments carried out on the cultivar ‘Hayward’ with deltamethrin (#5, #6), the trumpets atomizer provided a significantly better performance compared to the conventional ray atomizer across the whole study period (Table 3). On average (considering together experiments #5 and #6), in the final assessment the trumpets-modified atomizer caused an insect mortality 1.4-fold and 2.4-fold higher than the conventional ray atomizer and the untreated control, respectively.

### 3.2. Effect of Insecticide Spary Technique on Fruit Damage Caused by Halyomorpha halys Natural Infestation

On average, the fruit damage incidence was 33.0 ± 8.9% in the untreated control, 15.7 ± 7.6% in the plots sprayed using the conventional ray atomizer, and 12.0 ± 1.0% in the plots sprayed with the trumpets-modified atomizer for the cultivar ‘Jintao’. In the cultivar ‘Hayward’, the fruit injury level reached 46.7 ± 7.0% in the untreated control, 15.0 ± 3.6% in the plots sprayed using the conventional ray atomizer, and 16.7 ± 2.1% in the plots sprayed with the trumpets-modified atomizer. The fruit injury severity reflected the fruit injury incidence in both the cultivars under study. The factorial ANOVA carried out on log-transformed mean number of *H. halys* feeding punctures per fruit detected significant main effects of both treatment and kiwifruit cultivar, but a lack of the interaction “treatment × cultivar“ (F_(2; 12)_ = 0.29; *p* = 0.75). A higher level of fruit injury was found in the untreated plots than in sprayed ones (F_(2; 12)_ = 23.71; *p* < 0.001), while no significant differences were detected between the insecticide spray techniques. The effect of the kiwifruit cultivar was also significant (F_(1; 12)_ = 29.87; *p* < 0.001), with a higher level of fruit injury in the cultivar ‘Hayward’ compared to the cultivar ‘Jintao’ (Figure 5).

### 3.3. Actinidia Chinensis Biometric Plant Parameters

The cultivar ‘Jintao’ always had 10 shoots per plant, while the cultivar ‘Hayward’ had 15–16 shoots per plant (15.8 ± 0.4). The average productive shoot length was 114.5 ± 9.9 cm and 165.7 ± 10.4 cm for the cultivars ‘Jintao’ and ‘Hayward’, respectively. The average number of leaves per shoot recorded prior to harvest was 40.4 ± 9.8 and 50.7 ± 7.7 for the cultivars ‘Jintao’ and ‘Hayward’, respectively. Finally, for the cultivar ‘Jintao’, the mean leaf area was 143.0 ± 74.6 cm^2^ and total leaf area for a shoot was 4820.3 ± 823.9 cm^2^. In the cultivar ‘Hayward’, the mean leaf area was 182.6 ± 68.4 cm^2^ and the total leaf area in a shoot was 8784.6 ± 1798.9 cm^2^. Considering these measured parameters, the total leaf area per plant, calculated according to the average number of shoots and leaves per each cultivar, could be estimated in 48,203 cm^2^ for ‘Jintao’ and 138,797 cm^2^ for ‘Hayward’ cultivars, respectively.

## 4. Discussion

The insect mortality recorded in this study testing *H. halys* adults confined in mesh cages was in line with previous data collected with the same experimental design [43] and higher than what was observed with a natural infestation in open field [50,51]. The insects received a topical insecticide application and were forced to stay in a contaminated environment; they were, therefore, fully exposed to the plant protection products under study. This experimental approach may lead to an overestimation of the real insect mortality, but allows us to evaluate the relative efficacy of the treatments under study with a comparable experimental set up [49].

Considering the insect mortality recorded with the two spray techniques under study, in the cultivar ‘Hayward’, the trumpet atomizer resulted in a higher efficiency than the conventional ray atomizer. However, in the cultivar ‘Jintao’, no difference emerged between the spray techniques. This result could be explained by the leaf area and canopy density in this latter cultivar. In fact, in comparison with ‘Hayward’, ‘Jintao’ had a lower number of shoots, which were shorter and with less leaves. In addition, the cultivar ‘Hayward’ had 1.3-fold wider leaves with an overall 2.9-fold higher total leaf area per plant compared to the cultivar ‘Jintao’. These differences in terms of canopy structure, shape, and size likely influenced the insecticide distribution using either the conventional ray atomizer or the trumpets-modified atomizer in the green-flesh cultivar ‘Hayward’. This effect was more evident using the synthetic pyrethroid deltamethrin, likely because of the higher insecticidal activity against *H. halys* compared to the natural pyrethrins [43,51].

According to the *H. halys* monitoring trap catches (Figure 4), the pest pressure started to increase from mid-September, reaching a peak of adults in the first half of October, 12 and 23 days prior to the harvest of ‘Jintao’ and ‘Hayward’ cultivar, respectively. The insecticide strategies provided a sufficient, but not fully satisfactory damage control, with an Abbott efficacy close to 60% for the cultivar ‘Jintao’ and close to 65% for ‘Hayward’ in terms of damage incidence reduction compared to the untreated control.

Overall, *H. halys* damage incidence was higher for the cultivar ‘Hayward’, and the green-flesh fruits were significantly more injured (with a higher damage severity) compared to the yellow-flesh fruits. It has to be noted that the cultivar ‘Hayward’ was harvested 11 days after the cultivar ‘Jintao’ and therefore remained exposed to the feeding activity of *H. halys* for a longer period. In addition, the ‘Hayward’ orchard was not covered with nets, resulting in a full exposure to the *H. halys* infestation that in the ‘Jintao’ orchard was limited by the anti-hail net combined with lateral nettings. Likely, the monoblock netting system usually applied in the yellow-flesh kiwifruit orchard helped to physically control *H. halys* [30,31,32,33]. Interestingly, these results are in line with Francati et al. [8], who showed how the kiwifruit injury increases across the season and is higher in late summer. The fruit injury results from our study do not allow us to discriminate a difference between the spray techniques, while both of them significantly reduced the fruit injury severity compared to the untreated control. Likely, other factors involved in this field study, including the high pest pressure recorded within the experimental site and the spray timing of the insecticides, could explain this result.

## 5. Conclusions

This study demonstrated how a localized spray technique specifically designed for kiwifruit can increase the effectiveness of insecticides application, especially in highly dense canopy crops such as in *A. chinensis* var. *deliciosa* ‘Hayward’. This trend was not consistent in the cultivar ‘Jintao’, likely due to the different canopy density and structure. Regardless of the plant density per hectare, which in this study was higher for the yellow-flesh cultivar compared to the green-flesh cultivar, the key factor involved in the performance of the spray technique was related to the canopy density per plant, considering variables such as the number of shoots, the shoots length, the number of leaves, and the leaf size. Despite a higher insect mortality measured with an artificial infestation of *H. halys*, the fruit injury level caused by the natural *H. halys* population resulted comparable between these spray techniques, regardless of the kiwifruit cultivar. Our results open a perspective to implement the chemical control of *H. halys* in kiwifruit orchards by refining the application techniques, in particular, localizing the insecticide distribution in the fruit area underneath the pergola training system in plantations where the plant canopy could act as a barrier for an optimal distribution of the plant protection products in the kiwifruit area. This approach should be tested also for other active substances, considering for instance that for kiwifruit the maximum residue level (MRL) of deltamethrin in Europe was recently reduced to 0.01 mg kg^−1^, resulting in an important limitation of use, with the suspension of deltamethrin-based sprays well before the pre-harvest interval (PHI) foreseen in the label of insecticide commercial formulations [52]. Further experiments will aim at measuring the effect of trumpet atomizer on the reduction in insecticide dose and spray frequency, as well as evaluating the side impacts of these spray techniques on non-target arthropods in field conditions.

## Figures and Tables

**Figure 1 insects-15-00723-f001:**
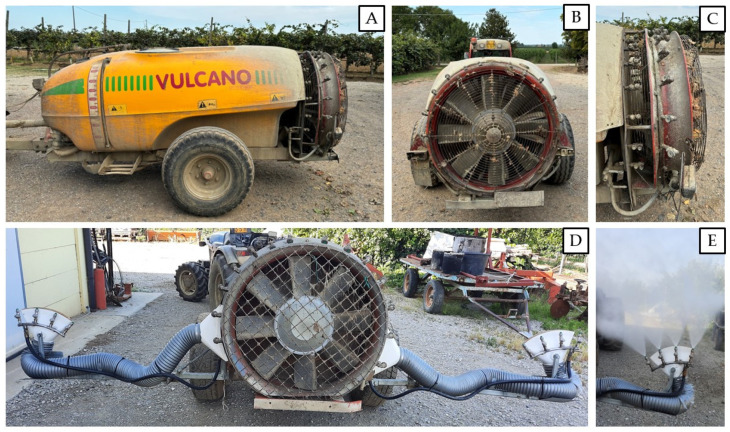
(**A**) Conventional ray atomizer commonly used by kiwifruit growers in Italy. (**B**,**C**) Detail of the nozzles installed radially around the fan. (**D**) Trumpets-modified atomizer used to localize the insecticides distribution in the fruit area. (**E**) Detail of the ‘trumpets’ installed in the localized spray technique and used in this study to apply insecticides against *Halyomorpha halys* in the kiwifruit orchard delivering the products in the fruit area.

**Figure 2 insects-15-00723-f002:**
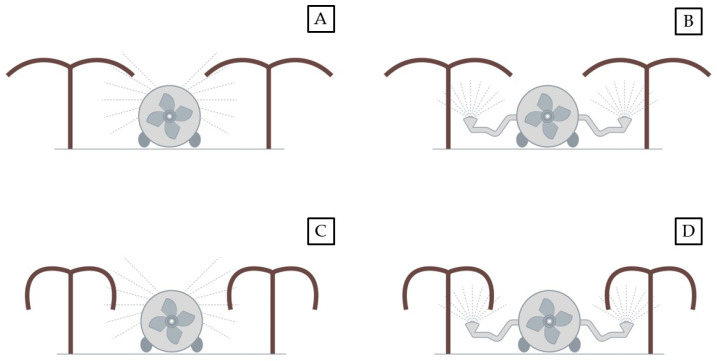
Schematic representation of kiwifruit vine training system ‘pergola’ for *Actinidia chinensis* var. *chinensis* ‘Jintao’ (**A**,**B**) and *A*. *chinensis* var. *deliciosa* ‘Hayward’ (**C**,**D**) with the conventional ray atomizer and the trumpets-modified atomizer, respectively.

**Figure 3 insects-15-00723-f003:**
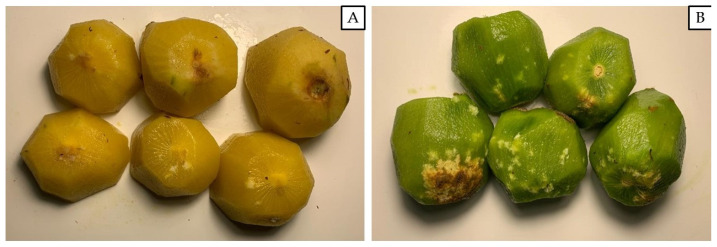
Kiwifruit with internal corking due to *Halyomorpha halys* trophic activity; each white spot corresponds to a *H. halys* feeding puncture. (**A**). Damage on *Actinidia chinensis* var. *chinensis* kiwifruit cultivar ‘Jintao’. (**B**). Damage on *A. chinensis* var. *deliciosa* cultivar ‘Hayward’.

**Figure 4 insects-15-00723-f004:**
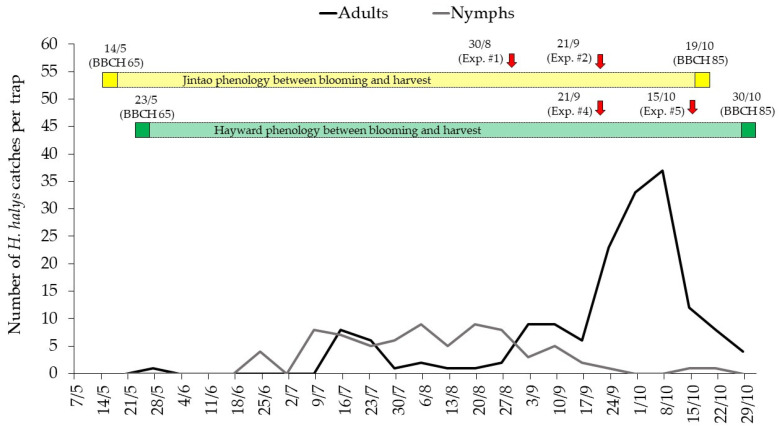
Trend of *Halyomorpha halys* catches during season 2021 in the Dead-Inn^®^ pyramid monitoring trap (AgBio Inc. Westminster, CO, USA) baited with Pherocon^®^ BMSB lure (Trécé Inc. Adair, OK, USA). The kiwifruit phenology between blooming (crop BBCH 65) and harvest (crop BBCH 85) is reported for cultivars ‘Jintao’ (in yellow) and ‘Hayward’ (in green) together with the insecticide applications carried out in 2021.

**Figure 5 insects-15-00723-f005:**
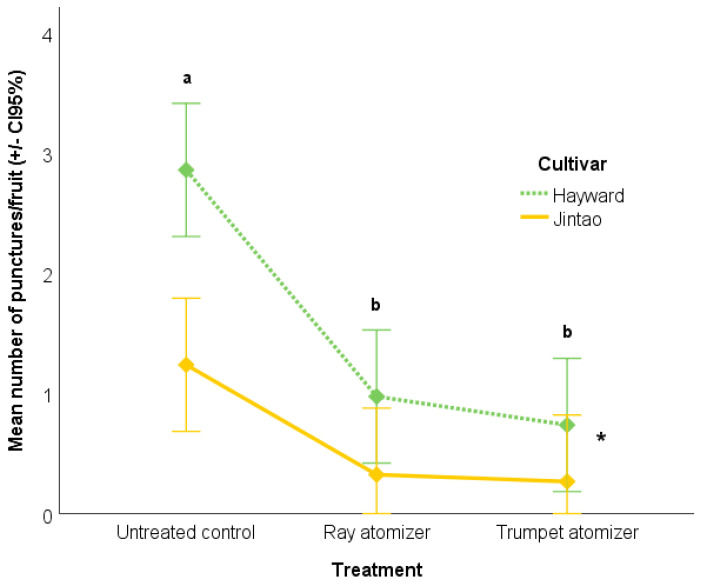
Effects of treatments and kiwifruit cultivars on the mean number of *H. halys* feeding punctures per fruit. Different letters indicate a significant difference among the treatments as detected by the Tukey HSD test (*p* < 0.05), and the asterisk indicates a significant main effect of the kiwifruit cultivars.

**Table 1 insects-15-00723-t001:** Outputs of the GLMMs with binomial error distribution and probit link function testing the effects of treatment, days of exposure and the interaction “treatment × days of exposure” on the mortality of *Halyomorpha halys* adults exposed within mesh cages in kiwifruit orchard of cultivar ‘Jintao’ (experiments #2 and #3) and cultivar ‘Hayward’ (experiments #4, #5, and #6).

Experiment	Factor	F	df_1_	df_2_	*p* Value
#2 Pyrethrins, 21 September 2021	Treatment	3.61	2	150.58	0.03
	Days of exposure	48.97	3	276.83	<0.001
	Treatment × Days of exposure	3.43	6	273.97	0.002
#3 Deltamethrin, 26 September 2022	Treatment	44.17	2	94.20	<0.001
	Days of exposure	44.19	3	264.86	<0.001
	Treatment × Days of exposure	3.12	6	267.40	0.01
#4 Pyrethrins, 21 September 2021	Treatment	12.80	2	95.63	<0.001
	Days of exposure	49.39	3	254.10	<0.001
	Treatment × Days of exposure	0.98	6	259.59	0.44
#5 Deltamethrin, 15 October 2021	Treatment	40.42	2	95.79	<0.001
	Days of exposure	22.04	3	279.82	<0.001
	Treatment × Days of exposure	4.13	6	277.14	<0.001
#6 Deltamethrin, 26 September 2022	Treatment	34.99	2	112.15	<0.001
	Days of exposure	8.77	3	272.35	<0.001
	Treatment × Days of exposure	1.92	6	280.90	0.08

**Table 2 insects-15-00723-t002:** Mean percent mortality (95% confidence interval lower and upper bound) of *Halyomorpha halys* adults confined in mesh cages recorded at 1, 3, 7 and 10 days after insecticide application with either deltamethrin or pyrethrins in kiwifruit orchard of cultivar ‘Jintao’ using either a conventional ray atomizer or trumpets-modified atomizer.

		Mean Percent Mortality (95% CI)			
Experiment	Treatment	1 Day	3 Days	7 Days	10 Days
#1 Deltamethrin, 30 August 2021	Control	26.7 (17.8–35.6)	-	-	-
	Conventional atomizer	60.0 (48.5–71.5)	-	-	-
	Trumpet atomizer	57.8 (44.7–70.8)	-	-	-
#2 * Pyrethrins, 21 September 2021	Control	0.0 (-) c	4.4 (0.1–8.7) c	28.9 (19.2–38.5) b	28.9 (19.2–38.5) b
	Conventional atomizer	27.8 (19.1–36.5) a	38.9 (29.6–48.2) a	60.0 (53.1–66.9) a	74.4 (65.4–83.5) a
	Trumpet atomizer	11.1 (4.3–17.9) b	23.3 (15.9–30.7) b	54.4 (44.9–64.0) a	74.4 (66.7–82.2) a
#3 * Deltamethrin, 26 September 2022	Control	2.2 (0.0–5.4) c	6.7 (1.6–11.7) c	24.4 (14.1–34.7) b	43.3 (29.8–56.9) b
	Conventional atomizer	33.3 (21.5–45.1) b	50.0 (38.8–61.2) b	83.3 (76.2–90.5) a	90.0 (84.2–95.8) a
	Trumpet atomizer	67.8 (55.8–79.8) a	67.8 (55.8–79.8) a	82.2 (74.4–90.0) a	90.0 (84.2–95.8) a

*: Because of the significant effect of the interaction “treatment × days of exposure” (Table 1), treatments were separated within each exposure interval by sequential Bonferroni method. Different letters indicate significant differences (*p* < 0.05) among treatments within each date.

**Table 3 insects-15-00723-t003:** Mean percent mortality (95% confidence interval lower and upper bound) of *Halyomorpha halys* adults confined in the mesh cages recorded at 1, 3, 7, and 10 days after insecticide application with either pyrethrins or deltamethrin in kiwifruit orchard of cultivar ‘Hayward’ using either a conventional ray atomizer or trumpets-modified atomizer.

		Mean Percent Mortality (95% CI)			
Experiment	Treatment	1 Day	3 Days	7 Days	10 Days
#4 † Pyrethrins, 21 September 2021	Control (b)	2.2 (0.0–5.4)	11.1 (5.1–17.1)	21.1 (12.8–29.4)	37.8 (26.1–49.4)
	Conventional atomizer (a)	11.1 (3.6–18.7)	23.3 (13.4–33.2)	45.6 (35.5–55.6)	57.8 (48.0–67.5)
	Trumpet atomizer (a)	15.6 (7.7–23.4)	30.0 (21.8–38.2)	46.7 (38.9–54.4)	72.2 (63.5–80.9)
#5 * Deltamethrin, 15 October 2021	Control	8.9 (1.6–16.1) c	18.9 (7.2–30.5) c	25.6 (12.6–38.5) c	36.7 (22.3–51.0) c
	Conventional atomizer	43.3 (32.3–54.2) b	47.8 (36.6–59.0) b	47.8 (36.6–59.0) b	58.9 (48.2–69.6) b
	Trumpet atomizer	81.1 (73.3–88.9) a	85.6 (78.5–92.6) a	93.3 (88.3–98.4) a	94.4 (89.7–99.2) a
#6 † Deltamethrin, 26 September 2022	Control (c)	4.4 (0.1–8.7)	12.2 (6.1–18.3)	22.2 (12.2–32.2)	45.6 (34.5–56.6)
	Conventional atomizer (b)	46.7 (33.0–60.4)	56.7 (43.9–69.4)	71.1 (59.9–82.3)	77.8 (68.3–87.2)
	Trumpet atomizer (a)	83.3 (74.2–92.4)	94.4 (89.7–99.2)	100.0 (-)	100.0 (-)

*: Because of the significant effect of the interaction “treatment × days of exposure” (Table 1), treatments were separated within each exposure interval by the sequential Bonferroni method. Different letters indicate significant differences (*p* < 0.05) among treatments within each date. †: Only significant effects of the main factors were detected by GLMMs (Table 1) and different letters indicate significant differences (*p* < 0.05) among treatments as detected by the sequential Bonferroni method.

## Data Availability

Raw data of this article were uploaded in the public repository Zenodo and are available at this link: https://zenodo.org/records/13258508. https://doi.org/10.5281/zenodo.13258507.

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
