# Peer review of "Effect of the Localized Insecticides Spray Technique to Control Halyomorpha halys in Actinidia chinensis Orchards"

_insects, 2024, doi:10.3390/insects15090723_

Round 1

Reviewer 1 Report

Comments and Suggestions for Authors

Abstract should give some more clear results of the efficacy, in this form is similar to simple abstract.

Introduction is well prepared covering broad area of host cultivation and HH importance. However, some data showing more clear picture on HH biological cycle and seasonal activity from the kiwi fruit set up till harvest time would benefit a lot for the readers. 

Methodology is well described, and flow rate is similar same as water distribution. Still, if the main difference between these spray machineries is in the area where the air flow exits, do authors have prove that coverage of the plant by 2 system is same, using for example water sensitive tapes, or just comparing 2 machinery tools.

Line 222, active ingredient is piretrin, check is the pyrethtins if right termin

Figure 4 is fine but it fits to the Result section as the justification of the susceptible period.

results: 393 - if this is deltametrin, would be easier to understand

Author Response

Dear Editor and Reviewers,

we are pleased to receive your positive feedback regarding the manuscript ID titled ‘Effect of the localized insecticides spray technique to control Halyomorpha halys in Actinidia chinensis orchards.’ and we are glad you have found it suitable for the publication in Insects.

We made an effort to improve both the manuscript content and form, following the suggestions provided by the two Reviewers. Here below, we reply point by point to each comment. Thanks for your time spent on this manuscript review and for the valuable comments provided.

We now hope that the manuscript in this new edited and updated form can be published in Insects.

Abstract should give some more clear results of the efficacy, in this form is similar to simple abstract.

We slightly edited the Simple Summary and we added more clear results in the Abstract to better differentiate these two sections.

Introduction is well prepared covering broad area of host cultivation and HH importance. However, some data showing more clear picture on HH biological cycle and seasonal activity from the kiwi fruit set up till harvest time would benefit a lot for the readers. 

Thanks for the suggestion. Added a paragraph (see L79-85).

Methodology is well described, and flow rate is similar same as water distribution. Still, if the main difference between these spray machineries is in the area where the air flow exits, do authors have proven that coverage of the plant by 2 system is same, using for example water sensitive tapes, or just comparing 2 machinery tools.

We have performed preliminary tests with water-sensitive cards to evaluate the coverage with the two systems. We decided not to include these data in the present manuscript.

Line 222, active ingredient is piretrin, check is the pyrethtins if right termin

Checked. Pyrethrins is correct.

Figure 4 is fine but it fits to the Result section as the justification of the susceptible period.

We agree with you, it is part of the results supporting the insecticide sprays in the fruit injury study carried out during season 2021 with the natural pest infestation. Nevertheless, we prefer to keep Figure 4 here in order not to distract the reader from the mortality and damage results once in the results section.

results: 393 - if this is deltametrin, would be easier to understand

Agreed and edited adding the specification of the active substance.

Reviewer 2 Report

Comments and Suggestions for Authors

Brief Summary: This study compared two spray techniques, 1) a broadcast spray, facilitated by a conventional ray atomizer; and 2) a targeted spray, facilitated by a “trumpet”-modified ray atomizer; treatments were compared against an untreated control. The experiment was conducted in two neighboring orchard blocks, each planted with a different cultivar (yellow and green kiwifruit). The green kiwifruit block was lower density, did not have exclusion/hail netting, and was pruned/trellised so that the canopy was more concave relative to the spray angle achieved by the sprayers. The yellow kiwifruit block was higher density, had exclusion/hail netting, and was trellised/pruned so that the undersides of the canopies were less concave than in the green cultivar, relative to the angle of the sprayers. The results of this study indicate that the trumpeted modification to the ray atomizer may have merit for this system, especially under the conditions in the green cultivar, evidenced by differences in insect mortality in the ‘Hayward’ block when deltamethrin was applied. When pyrethrins were applied, the trumpet performed just as well as the conventional atomizer. In ‘Jianto’, it appears that, while coverage may have been more successfully targeted for the trumpet, evidenced by greater initial knockdown, it was not significantly different from the conventional spray technique, given that mortality was equivalent between treatments by 10 days. The authors acknowledge in their discussion that the similar mortality between treatments in ‘Jianto’ was likely due to insects being confined within contaminated screens. Considering this, the fact that differences in mortality were observed at 10-days in ‘Hayward’ points to success of the trumpet attachment to deliver a more targeted insecticide application.

General concept comments
Article:
This manuscript is well-written, has research objectives that are clearly outlined, and describes experimental procedures and analyses that are appropriate for addressing their research questions. The statistical procedures are sound, and conclusions are measured and appropriate based on the methodologies outlined.

Specific comments:

The introduction may be made stronger by mentioning the conventional spray equipment in a little more detail, specifically, the ray atomizer. The authors thoroughly cover why traditional spray equipment may be inefficient, but do not mention the atomizer until the Materials and Methods. This could be addressed in a line or two, perhaps in paragraph L104-116. I will not note this as a mandatory change, however, since the authors thoroughly explain the two equipment types under Materials and Methods.

L208-219: How far apart were plots?

The opening lines of the Conclusions section are a little confusing because the reader has been operating under the assumption that the ‘Hayward’ cultivar was lower density. Whether the ‘Hayward’ block is high density relative to what is common in kiwi production is not clear in the text. This renders the conclusion that a localized spray may increase the effectiveness of insecticides in high density plantings, a little confusing. While the authors make it clear that the ‘Hayward’ block is lower density relative to ‘Jianto,’ the authors should consider clarifying this further. (L417 - 419)

I cannot read the text in Figure 4. Authors should include figures with higher dpi, especially line graphs.

Comments on the Quality of English Language

The writing is generally good. I noticed one typo (L70, "sting bug"). There are a few areas where definite articles, such as "the" are used inappropriately or unnecessarily. Some examples are "The H. halys management" on L79, and "the chemical control of H. halys using insecticides" on L90. Generally, though, the writing is clear and the information is conveyed effectively, therefore requires no major changes.

Author Response

Dear Editor and Reviewers,

we are pleased to receive your positive feedback regarding the manuscript ID titled ‘Effect of the localized insecticides spray technique to control Halyomorpha halys in Actinidia chinensis orchards.’ and we are glad you have found it suitable for the publication in Insects.

We made an effort to improve both the manuscript content and form, following the suggestions provided by the two Reviewers. Here below, we reply point by point to each comment. Thanks for your time spent on this manuscript review and for the valuable comments provided.

We now hope that the manuscript in this new edited and updated form can be published in Insects.

General concept comments

Article: This manuscript is well-written, has research objectives that are clearly outlined, and describes experimental procedures and analyses that are appropriate for addressing their research questions. The statistical procedures are sound, and conclusions are measured and appropriate based on the methodologies outlined.

Thanks for this comment.

Specific comments:

The introduction may be made stronger by mentioning the conventional spray equipment in a little more detail, specifically, the ray atomizer. The authors thoroughly cover why traditional spray equipment may be inefficient, but do not mention the atomizer until the Materials and Methods. This could be addressed in a line or two, perhaps in paragraph L104-116. I will not note this as a mandatory change, however, since the authors thoroughly explain the two equipment types under Materials and Methods.

Thanks for this note. We prefer to avoid repetition and leave the specification of the spaying machineries in M&M. Nevertheless, we added a couple of sentences in that paragraph to highlight other important factors that require to be considered when spraying.

L208-219: How far apart were plots?

Info added, see L222. Plots were at least 25 m apart from each other.

The opening lines of the Conclusions section are a little confusing because the reader has been operating under the assumption that the ‘Hayward’ cultivar was lower density. Whether the ‘Hayward’ block is high density relative to what is common in kiwi production is not clear in the text. This renders the conclusion that a localized spray may increase the effectiveness of insecticides in high density plantings, a little confusing. While the authors make it clear that the ‘Hayward’ block is lower density relative to ‘Jianto,’ the authors should consider clarifying this further. (L417 - 419)

Thanks for pointing out this aspect. Regardless the plant density (which in this study is higher for the yellow-flesh cv ‘Jintao’), the key point is the canopy density considering the shoots length and foliar coverage, which is much higher for the green-flesh cv ‘Hayward’. We edited this part in order to highlight the key message, regardless the plant density.

I cannot read the text in Figure 4. Authors should include figures with higher dpi, especially line graphs.

Figure 4 has been edited and improved. We also slightly improved and therefore changed Figure 2.

Comments on the Quality of English Language

The writing is generally good. I noticed one typo (L70, "sting bug"). Thanks, corrected. There are a few areas where definite articles, such as "the" are used inappropriately or unnecessarily. Some examples are "The H. halys management" on L79, and "the chemical control of H. halys using insecticides" on L90. Generally, though, the writing is clear and the information is conveyed effectively, therefore requires no major changes.

We went across the text for a throughout editing. Thanks.
